# Transfusion Trends of Knee Arthroplasty in Korea: A Nationwide Study Using the Korean National Health Insurance Service Sample Data

**DOI:** 10.3390/ijerph19105982

**Published:** 2022-05-14

**Authors:** You-Sung Suh, Hyung-Suk Choi, Jeong Seok Lee, Byung-Woong Jang, Jinyeong Hwang, Min Gon Song, Jaeeun Joo, Haran Chung, Jeong Jae Lee, Jae-Hwi Nho

**Affiliations:** 1Department of Orthopaedic Surgery, Soonchunhyang University Seoul Hospital, 59 Daesagwan-ro, Yongsan-gu, Seoul 04401, Korea; yssuh@schmc.ac.kr (Y.-S.S.); knee@schmc.ac.kr (H.-S.C.); 124856@schmc.ac.kr (J.S.L.); 99845@schmc.ac.kr (B.-W.J.); 129779@schmc.ac.kr (J.H.); 129687@schmc.ac.kr (M.G.S.); 2Data Science Team, Hanmi Pharm. Co., Ltd., 550, Dongtangiheung-ro, Gyeonggi-do, Hwaseong-si 18469, Korea; jaeeun.joo@hanmi.co.kr; 3The Center for Bloodless Medicine and Surgery and Patient Blood Management, Soonchunhyang University Seoul Hospital, 59 Daesagwan-ro, Yongsan-gu, Seoul 04401, Korea; s4288@schmc.ac.kr (H.C.); jjlee@schmc.ac.kr (J.J.L.)

**Keywords:** transfusion, trend, knee, arthroplasty, national health insurance service, big data

## Abstract

Knee arthroplasties are strongly associated with blood transfusion to compensate for perioperative bleeding. The purpose of this study was to evaluate trends of transfusion associated with knee arthroplasties using nationwide data of the National Health Insurance Service-National Sample Cohort (NHIS-NSC). Using data from the nationwide claims database of the Health Insurance Review Assessment Service managed by the NHIS, 50,553 knee arthroplasties under three categories (total knee replacement arthroplasty, uni-knee replacement arthroplasty, and revision arthroplasty) from 2012 to 2018 were identified. Overall transfusion rate, transfusion count, proportion of each type of transfusion, and cost associated with each type of operation were investigated. Overall transfusion rate was 83.4% (5897/7066) in 2012, 82.7% (5793/7001) in 2013, 79.6% (5557/6978) in 2014, 75.9% (5742/7557) in 2015, 73.1% (6095/8337) in 2016, 68.2% (4187/6139) in 2017, and 64.6% (4271/6613) in 2018. The proportion of each type of transfusion was 1.8% for fresh frozen plasma, 0.5% for platelets, and 97.7% for red blood cells. The average cost of transfusion was $109.1 ($123 in 2012, $124 in 2013, $123.3 in 2014, $110.6 in 2015, $100 in 2016, $92.9 in 2017, and $90.1 in 2018). In this nationally representative study of trends in transfusion associated with knee arthroplasty, we observed significantly high rates of blood transfusion among patients undergoing knee arthroplasties. Although the overall rate of transfusion had declined, the allogeneic transfusion rate was still high from 2012 to 2018 in Korea. Thus, surgeons need to develop various patient blood management plans and minimize the use of allogeneic transfusion when performing knee arthroplasties.

## 1. Background

When performing joint arthroplasty, allogeneic transfusions are often required due to intraoperative and postoperative bleeding caused by the operation [1,2,3]. Especially, knee arthroplasties are highly associated with transfusion to compensate for perioperative bleeding [1,4,5,6,7]. Total knee replacement is one of the most common operations in orthopedic practice, and during the period 2011–2018, the cumulative patient number for knee arthroplasty increased from 44,361 to 64,456 and the patient rate per 100,000 people increased from 44.3 to 62.8 in Korea [8]. Approximately one-third of patients have been reported to require transfusion of one to three units of blood, although the range of reported transfusion rates is large (20% to 70%) [9]. Evaluation of transfusion trends by the Korean Health Insurance Review Agency has suggested that transfusion for joint arthroplasties accounts for 6.2~8% of the entire transfusion volume [7,10].

The prevalence of knee arthritis will be increased. Knee arthroplasties are performed more frequently with an increasing elderly population. Older people are more likely to have anemia, which often requires allogeneic blood transfusions in the perioperative period [11]. However, both medical staff performing the arthroplasty and patients are concerned about various problems caused by blood transfusion, including perioperative systemic or local infection, prolonged hospital stays, venous thromboembolic events, and mortality [1,3,4,5,6,11,12].

Recently, various attempts have been made to reduce allogeneic blood transfusion when performing arthroplasty. The transfusion rate is gradually decreasing with the development of knowledge about patient blood management [1,6,12,13,14]. However, few studies have investigated the transfusion trend of knee arthroplasty based on reliable big data. The Korean government operates a mandatory national health insurance system with a central database known as the National Health Insurance Service (NHIS). This database contains records of all prescription drugs and treatment claims for almost all Koreans. Thus, the objective of this study was to analyze current status and trends of transfusion (average transfusion rate and average transfusion cost of each operation) in knee arthroplasties using representative data from the NHIS in Korea involving the utilization of allogeneic blood transfusion for patients undergoing knee arthroplasty.

## 2. Methods

### 2.1. Subjects

The study design and protocol were approved by the Institutional Review Board of Soonchunhyang University Hospital, Korea. The National Health Insurance Service–National Sample Cohort (NHIS-NSC) is a population-based cohort established by the National Health Insurance Service (NHIS) and the Korean Health Insurance Review and Assessment Service (HIRA) in South Korea. The NHIS-NSC database is a population-based sample cohort. The sole purpose of constructing the cohort was to provide public health researchers and policymakers with representative, useful information regarding citizens’ utilization of health insurance and health examinations [15]. A total of 1,025,340 participants of the cohort accounting for 2.2% of the total eligible population were randomly sampled from the 2002 Korean (nationwide) health insurance database to obtain baseline data. Information about participants’ insurance eligibility, medical treatment history, healthcare provider’s institution, and general health examination are included [15].

The Korean NHIS provides health insurance to all Koreans. All Koreans are obliged to become members of this national insurance system. About 98% of the Korean population are included in this national insurance system. After data selection, the appropriateness of the sample was verified by a statistician who compared the data from the entire Korean population with the sample data. Details of methods used to perform these procedures are provided by the National Health Insurance Sharing Service. All clinics and hospitals submit claims data for inpatient and outpatient care, including diagnoses (using the International Classification of Diseases, 10th revision [ICD-10] codes), procedures distinguished by health insurance claim codes, prescription records, demographic personal information, mortality from the Korean National Statistical Office, and direct medical costs. Therefore, virtually all information about patients and their medical records are available. The NHIS-NSC database is known to represent Koreans effectively [7,15,16]. We used data from the nationwide claims database of the NHIS-NSC between 2012 and 2018. Most knee arthroplasties performed in Korea were covered by the NHIS from 2012 to 2018.

Data managed by the NHIS-NCS were used to identify 58,513 knee arthroplasties under three categories (total knee replacement arthroplasty (TKA), uni-knee replacement arthroplasty (UKA) or hemiarthroplasty, and revision arthroplasty (RA)) from 2012 to 2018. These three categories were classified using operation codes recorded in the claims data of the NHIS. Operation codes were N2072 and N2077 for total knee replacement arthroplasty, N2712 and N2717 for uni-knee replacement arthroplasty, and N4727, N4722, N4717, N4712, N3727, N3722, N3717, and N3712 for revision arthroplasty. Most of the claims data carried a single operation code. We excluded 8822 knee arthroplasties (15.07% of overall knee arthroplasties) with multiple operation codes (two or more operation codes for knee arthroplasty) in one claim to determine the precise need for transfusion. These 8822 cases had bilateral operations or reoperations for complications such as infection, peri-prosthetic fracture, malposition, or dislocation. After excluding patients with two or more operation codes, the remaining 49,691 patients who had only a single operation code in their claim data were enrolled in this study to obtain reasonable results (Figure 1).

After classifying operations into three categories (TKA, UKA, and RA), the presence of a transfusion code was identified to calculate transfusion information in the claim data. Transfusion codes were: X1001, X1002, X2011, and X2012 for whole blood; X2021, X2022, X2031, X2032, X2091, X2092, X2131, X2132, and X2512 for red blood cells (RBC); X2041, X2042, X2051, and X2052 for fresh frozen plasma (FFP); X2081, X2082, X2121, X2122, X2501, X2511, X2512, and X2513 for platelets; and other codes for cryoprecipitate, white blood cells, plasma, and other components.

Dependent variables were as follows: the presence or absence of blood transfusion (transfusion rate), transfusion amount, transfusion cost for each type of operation, and proportions of whole blood, fresh frozen plasma, platelets, red blood cells, and other components (cryoprecipitate, white blood cell, plasma, etc.) in the transfused blood.

### 2.2. Transfusion Rates, Amounts, Proportions, and Costs (Economic Burden)

To determine the type of blood transfusion in each operation, we confirmed the transfusion code along with the operation code in the claims data. The presence of more than one transfusion code with an operation code indicated a transfusion. However, not all transfusion codes were used. Transfusion codes related to knee arthroplasties were used only within 7 days before and after the operation code date. Transfusion codes appearing at more than 7 days from operation code dates were excluded because they were not directly related to a perioperative transfusion, and patients with more than 2 operation codes were excluded in one claim data.

The amount of transfusion was investigated by counting transfusion codes in claim data in each of these three groups. Transfusion codes in claim data for each group indicated unit counts of fresh frozen plasma (FFP), platelets (PLTs), and red blood cells (RBCs). The average amount of transfusion described was the sum of units associated with the transfusion. After identifying overall transfusion numbers and categories, the average amount of transfusion, the proportion of transfusion of RBCs, fresh frozen plasma, platelets, and other components were determined for knee arthroplasties. The cost of each operation and economic burden associated with transfusion were also investigated. All costs were calculated in US dollars (1$ = about 1124 won in 2012, 1094 won in 2013, 1053 won in 2014, 1132 won in 2015, 1161 won in 2016, 1130 won in 2017, and 1101 won in 2018).

### 2.3. Statistical Analysis

Baseline patient characteristics and perioperative clinical data were analyzed for all patients. Descriptive statistics were used for operation types. Total and average values were estimated for the count of transfusion and cost while frequencies and percentages were used for identifying the transfusion rate. To determine the linear trend of annual transfusion rates, we performed the linear-by-linear test. We performed Poisson regression analysis with a canonical link function to adjust for overdispersion. The relative risk ratio (RR) and 95% confidence interval (CI) are represented to show the effect size difference between operation type, age, gender, hospital type, and area. All tests were two-sided and statistical significance was accepted at *p*-value < 0.05. All statistical analyses were performed using SAS Enterprise Guide version 6.1 (SAS Institute Inc., Cary, NC, USA) and R version 3.4.1 (RStudio, Boston, MA, USA).

## 3. Results

### 3.1. Transfusion Rates

The proportion of patients who received allogeneic transfusions (transfusion rate) among patients who underwent knee arthroplasty was investigated. Transfusion rates were classified into three subtypes depending on the type of knee arthroplasty (total knee replacement arthroplasty, uni-knee replacement arthroplasty, and revision arthroplasty). From 2012 to 2018, the sample number of patients who underwent each knee arthroplasty in Korea and the ratio of each surgery were investigated using data from the NHIS to determine the percentage of patients treated with each knee arthroplasty who received a transfusion from 2012 to 2018 in Korea. The overall transfusion rate of allogeneic blood was 75.5% among total cases of knee arthroplasties performed from 2012 to 2018. The overall transfusion rate was 83.4% (5897/7066) in 2012, 82.7% (5793/7001) in 2013, 79.6% (5557/6978) in 2014, 75.9% (5742/7557) in 2015, 73.1% (6095/8337) in 2016, 68.2% (4187/6139) in 2017, and 64.5% (4271/6613) in 2018 (Table 1, Figure 2).

### 3.2. Transfusion Amounts and Transfusion Costs

The overall number of transfusions (total count) was investigated using each code in NHIS-NSC claim data. The total number of transfusions was 5897 in 2012, 5.793 in 2013, 5557 in 2014, 5742 in 2015, 6095 in 2016, 4187 in 2017, and 4271 in 2018 (Table 1).

The total cost of transfusion was calculated by multiplying the cost of each transfusion code and counts of the transfusion code in claims data. The average cost of transfusion was calculated by dividing the total cost of transfusion by the number of each operation code in NHIS-NSC claims data. The average cost of transfusion was $109.1 ($123 in 2012, $124 in 2013, $123.3 in 2014, $110.6 in 2015, $100 in 2016, $92.9 in 2017, and $90.1 in 2018). The annual average economic burden for transfusion in total knee replacement arthroplasty, uni-knee replacement arthroplasty, and revision arthroplasty are shown in Table 2, Table 3 and Table 4 and Figure 3.

**Table 1 ijerph-19-05982-t001:** Transfusion rates in knee arthroplasty classified according to the type of operation (total knee replacement arthroplasty, uni-knee replacement arthroplasty, and revision arthroplasty) and total transfusion rate in overall knee arthroplasties from 2012 to 2018 in Korea († *p*-value by linear-by-linear test).

	Transfusion	2012	2013	2014	2015	2016	2017	2018	Total	*p*-Value †
TKA	Transfused	5691	5576	5334	5507	5857	4045	4066	36,076	0.000
	Not transfused	861	876	1081	1417	1817	1597	1961	9610	
UKA	Transfused	115	117	115	125	118	80	86	756	0.000
	Not transfused	265	277	283	338	345	279	303	2090	
RA	Transfused	91	100	108	110	120	62	119	710	0.000
	Not transfused	43	55	57	60	80	76	78	449	
Total	Transfused	5897	5793	5557	5742	6095	4187	4271	37,542	0.000
	Not transfused	1169	1208	1421	1815	2242	1952	2342	12,149	
	Percentage (%)	83.46	82.75	79.64	75.98	73.11	68.20	64.58	75.55	
Subtotal		7066	7001	6978	7557	8337	6139	6613	49,691	
Multiple operation codes		825	1108	1237	1441	1614	1250	1347	8822	
Total count		7891	8109	8215	8998	9951	7389	7960	58,513	
Average cost		$123	$124	$123.3	$110.6	$100	$92.9	$90.1	$109.1	

TKA, total knee replacement arthroplasty; UKA, uni-knee replacement arthroplasty; RA, revision arthroplasty.

**Figure 2 ijerph-19-05982-f002:**
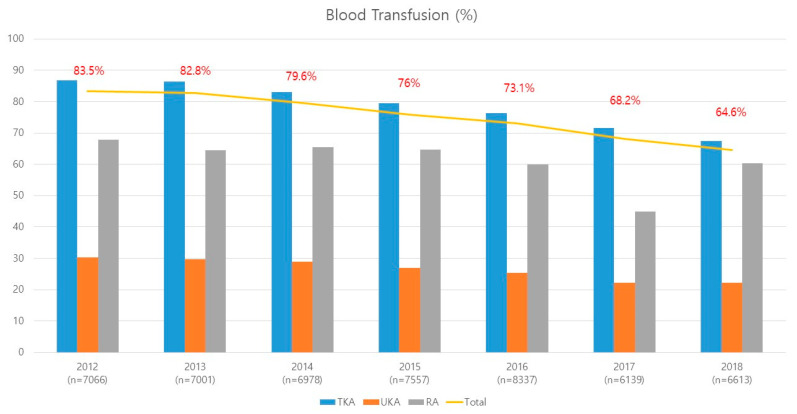
Transfusion rates in 49,691 knee arthroplasties from 2012 to 2018 in Korea. (TKA, total knee replacement arthroplasty; UKA, uni-knee replacement arthroplasty; RA, revision arthroplasty).

**Figure 3 ijerph-19-05982-f003:**
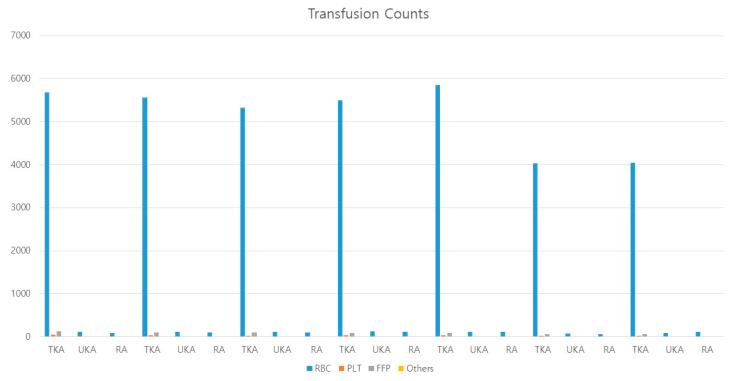
Overall and average counts of transfusion in TKA, UKA, and RA from 2012 to 2018 in Korea. (TKA, total knee replacement arthroplasty; UKRA, uni-knee replacement arthroplasty; RA, revision arthroplasty.

## 4. Discussion

As blood donation is decreased due to prolonged COVID-19, the Ministry of Health and Welfare (MOHW) in Korea has requested frontline medical institutions to establish an emergency blood management committee and start stable blood management. According to the blood management headquarters of the Korean Red Cross, the blood holding amount (red blood cell product) is 4.1 days, which is less than the optimal blood holding amount of 5 days. As of 14 August 2021, blood holdings are below the appropriate standard for all blood types. As instability of blood supply continues due to the continuous spread of COVID-19, the MOHW has recently sent an official letter to the Korean Hospital Association, requesting cooperation from medical institutions that use more than 1000 units of blood to establish a response plan according to the crisis stage of blood retention.

The current study showed transfusion rates and amounts for knee arthroplasties from 2012 to 2018 in Korea. This study using a nationwide claims database demonstrated that transfusion rates for knee arthroplasty declined slowly in Korea. However, they were still higher than those of other countries from 2012 to 2018. Nationwide data are important for evaluating the impact of allogeneic blood transfusion on health-related outcomes and the financial burden of knee arthroplasty. It is necessary to explain this from the point of view of a healthcare system. In Korea, National Health Insurance is implemented for almost all patients. Thus, every hospital charges the NHIS for medical expenses except for small expenses paid by the patient in this system. Therefore, blood transfusion status (rate, amount, cost) can be investigated very accurately using transfusion claim codes and operation claims recorded by the NHIS as well as on the NHIS-NSC database.

The NHIS-NSC database contains representative population-based cohort data. This is a major strength as this ensures its applicability in research. Moreover, its data are large-scale, extensive, and stable because they are constructed based on national health insurance data generated by the government or public institutions’ involvement [15].

Recent trends in transfusion rates and amounts of hip and knee arthroplasties in Korea suggest the need for new public health strategies. Analysis of costs associated with transfusion provides an insight into the economic burden associated with hip and knee arthroplasties, which in turn can guide the creation of public health policy [7].

During knee joint arthroplasty, bleeding cannot be avoided due to soft-tissue dissection and exposure of the hypervascular metaphysis after bone resection. Total joint arthroplasties are associated with substantial perioperative blood loss averaging 1000 to 2000 mL, with a decline in hemoglobin level [17,18,19]. With this blood loss, transfusion is performed due to a decrease of hemoglobin level associated with perioperative bleeding in knee arthroplasties. Several alternative transfusion methods are used to help a patient’s recovery from anemic condition, even if the actual perioperative blood loss is smaller than the expected blood loss, including iron supplement, recombinant erythropoietin, tranexamic acid, cell saver, etc. [1,3].

Allogeneic transfusion is restricted to reduce side effects of transfusion in most cases. Huge amounts of blood are still being used to correct anemia despite several complications such as intraoperative joint infection, systemic infection, internal organ damage, long-term hospitalization, venous thromboembolism, and death being reported [11,20,21,22]. Immunologically, allogeneic blood transfusions generally cause up-regulation of humoral immunity and down-regulation of macrophage and T-cell immunity. Furthermore, allogeneic transfusion can decrease the phagocytic activity of macrophages but increase glucocorticoid levels, resulting in suppression of the immune system [23,24,25]. Several observational studies have demonstrated associations of blood transfusion with increases of postoperative morbidity and mortality [26,27,28,29].

As the aging population increases, the number of patients who require knee joint replacement surgery is also increasing. Most patients undergoing knee arthroplasty are elderly patients who do not have sufficient ability to recover from intraoperative bleeding. Thus, postoperative blood transfusion is often necessary [1,4,6]. Furthermore, the blood donor population is decreasing in an elderly society despite an increasing need for blood [11]. Since the number of cases requiring blood transfusions continues to increase due to an increase in the elderly population, blood shortages and transfusion safety are emerging as social issues [1,6,30].

This study has several limitations. First, the exclusion of patients with multiple operation codes (15% of total claim data) might have influenced the overall results as it could lead to underestimated transfusion rates and amounts. Operation codes and transfusion codes were not distinguished in a patient with multiple operation codes. Second, the possibility of transfusion due to underlying diseases of patients cannot be excluded. This study only used transfusion codes related to knee arthroplasties within 7 days before and after the operation code date. However, not all transfusion codes about claims data were related to the operation. Thus, it is necessary to evaluate underlying diseases or conditions such as anemia, cancer, and other operations besides knee arthroplasty that need a transfusion. Third, the transfusion cost in this study was the actual cost of treatment for the patient recorded by the NHIS. Therefore, the cost can be estimated using transfusion codes recorded by the NHIS in this study. In the United States, there is a concern about the difference between blood acquisition costs and activity-based costs. It has been reported that activity-based costs of blood transfusions in surgical patients are 3.2- to 4.8-fold higher than blood product acquisition costs [31]. However, conceptual differences in transfusion costs exist in different health care systems. This study was designed based on nationwide claims data. Further analyses are needed to assess cost-effectiveness at individual patient, hospital, and societal levels. Advances in patient blood management and recent efforts to reduce transfusion may rapidly alter transfusion rates. Fourth, in this study, no additional analysis was performed according to age, gender, hospital type, or region. Fifth, the NHIS-NSC database has limitations. Disease codes listed in the cohort might not represent participants’ true disease status because the code was created to claim health insurance service for participants, an inherent limitation of insurance databases [15].

In summary, this nationally representative study of patterns of transfusion in knee arthroplasty revealed significantly high rates of blood transfusion among patients undergoing knee arthroplasties recently [1,4,17,21,22,32]. This is the first study to estimate the economic cost of blood transfusion due to knee arthroplasty in Korea. Although the total transfusion volume per patient decreased, there was no significant decrease in the transfusion rate in Korea from 2012 to 2018. The reason for the decrease in transfusion volume may be explained by advances in surgical techniques and the use of patient blood management. These transfusion rates in Korea were higher than those reported in other countries. Recently, efforts and interest in patient blood management are increasing in Korea [1,6,33]. Multiple factors may affect a surgeon’s decision for blood transfusion in knee arthroplasty. Thus, surgeons need to develop a patient blood management plan and minimize the use of allogeneic transfusion.

## 5. Conclusions

This national representative study on blood transfusion usage patterns in knee arthroplasty revealed that blood transfusion rates of patients undergoing knee arthroplasty in Korea were significantly higher than those in other countries. Thus, it is necessary to make an effort to reduce the transfusion rate using patient blood management.

This is the first study in Korea to estimate the blood transfusion rate and cost due to knee replacement arthroplasty. Although the overall transfusion rate is decreasing, perioperative allogeneic transfusion rates from 2012 to 2018 were still high. Thus, surgeons need to develop various patient blood management plans and minimize the use of allogeneic transfusion when performing knee arthroplasties.

## Figures and Tables

**Figure 1 ijerph-19-05982-f001:**
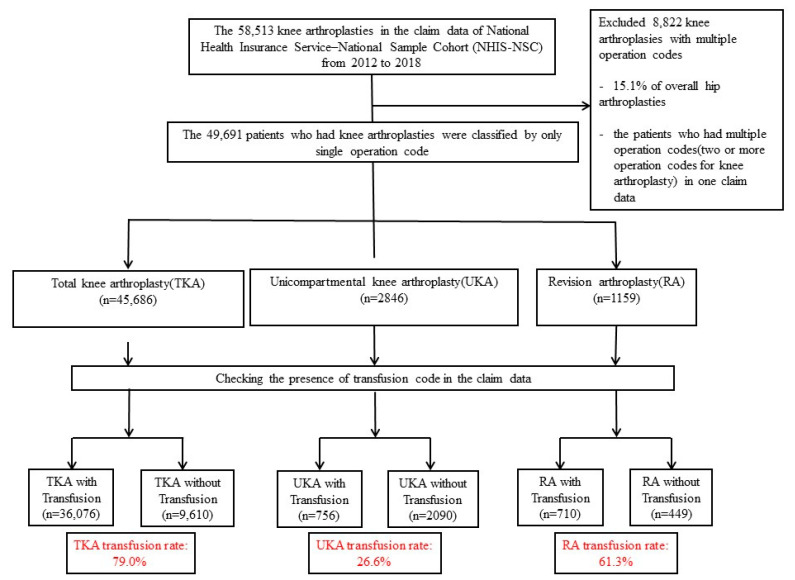
Flow chart showing the selection of knee arthroplasties in the claim data of the National Health Insurance Service. How to calculate the transfusion rate using data of the National Health Insurance Service (NHIS) from 2012 to 2018 in Korea is shown. Diagram also shows numbers of excluded, transfused, and non-transfused cases from hip arthroplasties. TKA: total knee arthroplasty; UKA: unicompartmental knee arthroplasty; RA: revision arthroplasty.

**Table 2 ijerph-19-05982-t002:** Overall amount and cost of transfusion in total knee replacement arthroplasty (TKA) from 2012 to 2018 in Korea.

Year	Number of Operations	Count of Transfusion Code	RBCs	PLTs	FFP	Others	Total
2012	6552	Total count	5680	47	128	11	5866
		Total cost	$723,040	$248	$1004	$4	$724,296
		Average cost	$127	$5	$8	$0.39	$141
2013	6452	Total count	5569	32	99	7	5707
		Total cost	$715,433	102	544	2	$716,081
		Average cost	$128	$3	$5	$0.31	$137
2014	6415	Total count	5325	25	99	12	5461
		Total cost	$675,466	$71	$564	$6	$676,107
		Average cost	$127	$3	$6	$0.50	$136
2015	6924	Total count	5498	31	93	13	5635
		Total cost	$622,231	$96	$478	$6	$622,811
		Average cost	$113	$3	$5	$0.45	$122
2016	7674	Total count	5857	38	90	7	5992
		Total cost	$585,740	$108	$306	$1	$586,155
		Average cost	$100	$3	$3	$0.15	$106
2017	5642	Total count	4038	23	58	7	4126
		Total cost	$372,154	$52	$180	$2	$372,388
		Average cost	$92	$2	$3	$0.26	$98
2018	6027	Total count	4051	22	57	5	4135
		Total cost	$358,552	$59	$148	$1	$358,760
		Average cost	$89	$3	$3	$0.28	$94

**Table 3 ijerph-19-05982-t003:** Overall amount and cost of transfusion in uni-knee replacement arthroplasty (UKA) from 2012 to 2018 in Korea.

Year	Number of Operations	Count of Transfusion Code	RBCs	PLTs	FFP	Others	Total
2012	380	Total count	114	1	1	0	116
		Total cost	$2634	$3	$1	$0	$2638
		Average cost	$23	$3	$1	$0	$27
2013	394	Total count	117	0	1	0	118
		Total cost	$2412	$0	$0	$0	$2412
		Average cost	$21	$0	$0	$0	$21
2014	398	Total count	111	3	4	1	119
		Total cost	$2452	$10	$4	$0	$2467
		Average cost	$22	$3	$1	$0.41	$27
2015	463	Total count	124	2	2	0	128
		Total cost	$2865	$6	$8	$0	$2878
		Average cost	$23	$3	$4	$0	$30
2016	463	Total count	117	2	1	0	120
		Total cost	$2207	$5	$1	$0	$2213
		Average cost	$19	$2	$1	$0	$22
2017	359	Total count	79	1	4	0	84
		Total cost	$1269	$2	$8	$0	$1280
		Average cost	$16	$2	$2	$0	$20
2018	389	Total count	86	0	0	0	86
		Total cost	$1368	$0	$0	$0	$1368
		Average cost	$16	$0	$0	$0	$16

**Table 4 ijerph-19-05982-t004:** Overall amount and cost of transfusion in revision arthroplasty (RA) from 2012 to 2018 in Korea.

Year	Number of Operations	Count of Transfusion Code	RBCs	PLTs	FFP	Others	Total
2012	134	Total count	90	2	13	1	106
		Total cost	$12,096	$85	$938	$2	$13,120
		Average cost	$134	$42	$72	$2	$251
2013	155	Total count	100	4	7	1	112
		Total cost	$14,088	$149	$368	$7	$14,612
		Average cost	$141	$37	$53	$7	$238
2014	165	Total count	108	3	4	1	116
		Total cost	$11,298	$38	$65	$1	$11,402
		Average cost	$105	$13	$16	$1	$135
2015	170	Total count	109	2	7	1	119
		Total cost	$9618	$11	$69	$2	$9700
		Average cost	$88	$6	$10	$2	$106
2016	200	Total count	120	3	9	0	132
		Total cost	$9859	$22	$180	$0	$10,061
		Average cost	$82	$7	$20	$0	$110
2017	138	Total count	61	1	6	0	68
		Total cost	$3863	$2	$96	$0	$3961
		Average cost	$63	$2	$16	$0	$81
2018	197	Total count	119	3	10	0	132
		Total cost	$12,176	$49	$185	$0	$12,410
		Average cost	$102	$16	$19	$0	$137

## Data Availability

The data that support the findings of this study are available from the National Health Insurance Service (NHIS) of Korea. Restrictions may apply to the availability of these data, which were used under license for the current study. These data are not publicly available. However, data are available from the authors upon reasonable request with permission of the National Health Insurance Service (NHIS) of Korea.

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
