# Peer review of "Transfusion Trends of Knee Arthroplasty in Korea: A Nationwide Study Using the Korean National Health Insurance Service Sample Data"

_ijerph, 2022, doi:10.3390/ijerph19105982_

Round 1

Reviewer 1 Report

Thank you for your submission. This was an interesting article to read and gratifying to see the steady decrease in transfusion rates. I have only a few comments to make:

  • (p8 line 214/215) it would be interesting to read the rates in other countries. You touched on the UK experience but it would be interesting to see how Korea fits into the world in this.
  • (p9 line 236) you mention other strategies are used to reduce allogeneic transfusion. I'm guessing these include iron supplementation, cell salvage, etc but it would be interesting to learn what you are actually doing
  • (p10 line 284) I was a bit confused by your statement that there was no significant change in transfusion rates. Were you referring to overall rates or in relation to knee surgery, because you've shown they are falling for knee surgery
  • (references:) consistent use of doi would be helpful for readers wanting to do further study on your work

Author Response

(p8 line 214/215) it would be interesting to read the rates in other countries. You touched on the UK experience but it would be interesting to see how Korea fits into the world in this.

Ans: Thank you for your detailed review. As suggested, I cited one of the IJERPH paper that explains the TKA trends in Korea.

“and over the 2011–2018, the cumulative patient number for knee arthroplasty increased from 44,361 to 64,456 and the patient rate per 100,000 people increased from 44.3 to 62.8 in Korea.[8]”

 (p9 line 236) you mention other strategies are used to reduce allogeneic transfusion. I'm guessing these include iron supplementation, cell salvage, etc but it would be interesting to learn what you are actually doing

Ans: Thank you for your kind recommendation. As you suggested, we changed.

“Several alternative transfusion methods are used to help patient’s recovery from anemic condition, even if the actual perioperative blood loss is smaller than the expected blood loss, including iron supplement, recombinant erythropoietin, and cell saver, etc.”

(p10 line 284) I was a bit confused by your statement that there was no significant change in transfusion rates. Were you referring to overall rates or in relation to knee surgery, because you've shown they are falling for knee surgery

Ans: Thank you for your point. To address your concerns, we have modified the following:

“Although the total transfusion volume per patient decreased, there was no significant change decrease in the transfusion rate in Korea from 2012 to 2018.”

(references:) consistent use of doi would be helpful for readers wanting to do further study on your work

Ans: I really appreciate your recommendation.

Reviewer 2 Report

General comment

I think this is an interesting paper attempting to evaluate Transfusion Trends of Knee Arthroplasty in Korea: A Nationwide Study Using the Korean National Health Insurance Service Sample Data. This study is worthy to readers. However, there are some issues which can be identified.

1) cause of decreasing trends of blood transfusion

The authors would better to describe the reason of decreasing trends of transfusion. For example, change of surgical technique, activation of patient blood management, and so on.

2) how to exclude simultaneous bilateral TKRA and very short interval of both TKRA within one or two weeks. .

Between Simple TKRA and bilateral TKRA are different condition for preventing transfusion. Therefore, this different surgical condition should be excluded or explain.

.  .

3) Dose use of pharmacologic agents, for example tranexamic acid, also access ?

In this big data might be include use of pharmacologic agent such as tranexamic acid. Is there any related with decreasing trend of transfusion in patients undergoing TKRA?

Introduction.

Well described

Patients and methods

Well described

Results

Well described

Discussion section

Please describe reason of decreasing trends of transfusion after TKRA.   

Author Response

General comment

I think this is an interesting paper attempting to evaluate Transfusion Trends of Knee Arthroplasty in Korea: A Nationwide Study Using the Korean National Health Insurance Service Sample Data. This study is worthy to readers. However, there are some issues which can be identified.

Ans: I really appreciate your valuable comments. I am going to explain point by point from the next paragraph, including your worries what you pointed out.

1) cause of decreasing trends of blood transfusion

The authors would better to describe the reason of decreasing trends of transfusion. For example, change of surgical technique, activation of patient blood management, and so on.

Ans: As you suggested, we added this sentence in the discussion to clarify.

“The reason for the decrease in transfusion volume may be explained by advances in sur-gical techniques and the use of patient blood management.”

2) how to exclude simultaneous bilateral TKRA and very short interval of both TKRA within one or two weeks. .

Between Simple TKRA and bilateral TKRA are different condition for preventing transfusion. Therefore, this different surgical condition should be excluded or explain.

Ans: Thank you for your kind comment. To avoid a point of concern, we have included only cases with one operation code in one claim data. Patients with more than 2 operation codes were excluded. Subsequent operations were excluded because it may affect anemia and transfusion.

3) Dose use of pharmacologic agents, for example tranexamic acid, also access ?

In this big data might be include use of pharmacologic agent such as tranexamic acid. Is there any related with decreasing trend of transfusion in patients undergoing TKRA?

Ans: Thank you for your kind comment. I agree with your opinion, but code extraction and retrieval for TXA was not conducted in this study. We will try it later. thank you.

Introduction.

Well described

Patients and methods

Well described

Results

Well described

Discussion section

Please describe reason of decreasing trends of transfusion after TKRA.  

Ans: I really appreciate your recommendation. As you suggested we added the reason in the discussion.

Reviewer 3 Report

This study was well conducted and is suitable for publication in the journal.

However, it needs meticulous English editing.   

Author Response

This study was well conducted and is suitable for publication in the journal.

However, it needs meticulous English editing.

Ans: Thank you for your kind comments. An additional certificate of editing for English proofreading is attached.
